# Burst muscle performance predicts the speed, acceleration, and turning performance of Anna's hummingbirds

Paolo S Segre[1][†], Roslyn Dakin[1], Victor B Zordan[2], Michael H Dickinson[3], Andrew D Straw[3,4], Douglas L Altshuler[1]*

[1]Department of Zoology, University of British Columbia, Vancouver, Canada; [2]Department of Computer Science and Engineering, University of California, Riverside, Riverside, United States; [3]Biology and Bioengineering, California Institute of Technology, Pasadena, United States; [4]Institute of Molecular Pathology, Vienna, Austria

**Abstract** Despite recent advances in the study of animal flight, the biomechanical determinants of maneuverability are poorly understood. It is thought that maneuverability may be influenced by intrinsic body mass and wing morphology, and by physiological muscle capacity, but this hypothesis has not yet been evaluated because it requires tracking a large number of free flight maneuvers from known individuals. We used an automated tracking system to record flight sequences from 20 Anna's hummingbirds flying solo and in competition in a large chamber. We found that burst muscle capacity predicted most performance metrics. Hummingbirds with higher burst capacity flew with faster velocities, accelerations, and rotations, and they used more demanding complex turns. In contrast, body mass did not predict variation in maneuvering performance, and wing morphology predicted only the use of arcing turns and high centripetal accelerations. Collectively, our results indicate that burst muscle capacity is a key predictor of maneuverability.

*For correspondence: doug@zoology.ubc.ca

Present address: [†]Hopkins Marine Station, Stanford University, Pacific Grove, United States

Competing interests: The authors declare that no competing interests exist

## Introduction

The ability of an animal to change the speed and direction of movement, defined as maneuverability (*Dudley, 2002*), can determine its success at avoiding predators, obtaining food, and performing other behaviors that determine the margin between life and death (*Webb, 1976*; *Hedenström and Rosén, 2001*; *Walker et al., 2005*). Most biomechanical research on birds has focused on either brief (e.g., take off) or steady state movements (e.g., forward flight) that can be studied most readily in the laboratory. Maneuverability is therefore one of the most important but least understood aspects of animal locomotion. Warrick and coworkers (*Warrick et al., 1988*; *Warrick and Dial, 1998*) proposed that there are both intrinsic and facultative influences on maneuvering performance. For animals that perform powered flight, intrinsic maneuverability is defined by the physical limitations imposed by morphology (*Norberg and Rayner, 1987*), but excess muscle capacity should allow them to facultatively overcome the costs of suboptimal morphology, achieving higher levels of performance by sacrificing efficiency. Although compelling, this hypothesis has never been tested explicitly.

Wing morphology is defined using measures of size (e.g., area or length) and non-dimensional measures of shape (e.g., aspect ratio). Wing area and aspect ratio have significant and well known effects on the aerodynamics of flight in animals (*Pennycuick, 1975*; *Kruyt et al., 2014*; *2015*), and should affect maneuvering performance. Wing morphology influences flight efficiency

**eLife digest** The ability of an animal to maneuver can determine its success at avoiding predators, catching prey, and outperforming its competitors. However, little is known about the characteristics that determine maneuverability. Why are some individuals more maneuverable than others?

To investigate this question, Segre et al. used an automated video tracking system to track male Anna's hummingbirds as they flew around a large chamber. These tracks were then compared with the physical characteristics of the birds to see which, if any, affect the birds' maneuverability. This revealed that body size did not affect how well the birds could maneuver. Instead, the muscle capacity of the birds – their ability to generate force rapidly – determined how well the birds performed most types of movement. Birds with higher muscle capacity flew faster, had faster accelerations and decelerations, could rotate their bodies more quickly, and performed more demanding and complex turns.

Segre et al. also found that wing shape is important for a type of maneuver called an arcing turn. Hummingbirds with a more slender wing shape were able to execute more demanding arcing turns involving higher accelerations, and they used arcing turns more often than birds with wider wings. Future research will aim to determine whether these relationships are also found in other species of birds.

(*Feinsinger and Chaplin, 1975*), ecological roles (*Feinsinger, 1976*; *Feinsinger and Colwell, 1978*; *Warrick, 1998*) and competitive ability (*Feinsinger and Chaplin, 1975*; *Feinsinger and Colwell, 1978*; *Feinsinger et al., 1979*; *Altshuler, 2006*). Because these previous studies focused on species and gender comparisons, less is known about how individual variation in wing morphology influences performance, especially with respect to maneuverability. One complication is that different wing sizes and shapes can be favored depending on the specific maneuver performed, e.g., yaw versus banked turns. Given the diversity of flight behaviors, it is unclear if the requirements for maneuvering exert strong selection on wing morphology.

Muscle capacity affects the maximum aerodynamic force a flying animal can produce. Aerodynamic force can be directed for performing maneuvers that require greater output than the minimum requirements for flight. Excess muscle capacity can also be used to compensate for anatomical or spatial constraints on wing movement (*Warrick, 1998*). Muscle output of hummingbirds has been quantified in several ways including oxygen consumption to determine metabolic input, wingbeat kinematics to estimate mechanical power output, and electromyography (EMG) to measure myoelectric input. Considering hovering flight as the point of comparison, forward flight at the fastest speeds recorded in a wind tunnel requires about 20% more metabolic (*Clark and Dudley, 2010*) and myoelectric input (*Tobalske et al., 2010*). Maximum sustained hovering performance has been studied by experimentally lowering air density to the lowest values in which birds are still able to hover. These experiments revealed that hovering in hypodense air requires ~40% higher mechanical power output (*Chai and Dudley, 1995*) and ~60% higher spatial recruitment of muscle fibers, as measured by the spike amplitude of the electromyogram recordings (*Altshuler et al., 2010b*), in comparison to hovering in normal air. By far the most expensive flight behavior studied to date in hummingbirds is maximum load lifting, which requires 200–400% more mechanical power output (*Chai et al., 1997*; *Chai and Millard, 1997*; *Altshuler et al., 2010a*), about 200% more spatial recruitment (EMG spike amplitude), and 150% more temporal recruitment (EMG spike frequency) (*Altshuler et al., 2010b*) compared to hovering.

Maximum load lifting is a transient behavior that uses the bird's natural escape response to measure burst power output. Thus, it is not surprising that this assay provides the maximum muscle capacity that has been measured in hummingbirds. It is particularly useful for quantifying variation among and within species in burst muscle capacity. Studies using the load lifting assay have revealed that maximum burst muscle capacity is related to hummingbird evolutionary ecology. Altshuler and coworkers (*Altshuler et al., 2004b*; *Altshuler, 2006*) demonstrated that ecological role is more strongly related to load lifting ability than morphological parameters such as wing loading. Load

lifting ability is also associated with species- and gender-specific competitive ability at different elevations. Altshuler (*Altshuler, 2006*) suggested that the relationship between maximum muscle capacity and competitive ability may be mediated through maneuvering performance.

Unconstrained maneuvering performance of birds, including hummingbirds, has recently been quantified in the field without individual identification (*Shelton et al., 2014*; *Sholtis et al., 2015*). Although field studies are valuable for quantifying average species performance, individual identification and large sample sizes are required to examine sources of within-species variation. Here, we studied the free-flight maneuvering performance of Anna's hummingbirds (*Calypte anna*) in a large flight chamber (*Video 1*). Flight maneuvers in a chamber are not expected to be the same as outdoors, and may have lower velocities and accelerations. The benefit of this approach is that a large number of measurements from the same individuals can be combined with other data to examine how variation in the observed maneuvers is influenced by individual morphology and muscle capacity.

We used a high-throughput computational approach to record the flight performance of 20

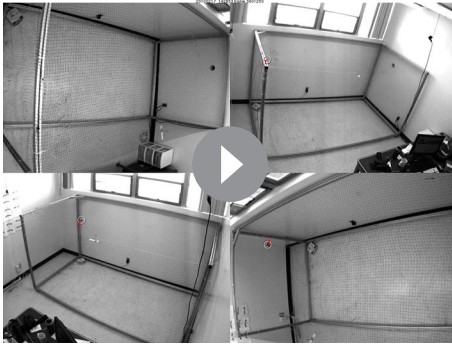

**Video 1.** The multi-camera, automated tracking system filming two hummingbirds in the flight arena at 200 frames per second. Continuously tracked sequences are assigned an object number (from 0 to 4 over this sequence). Body position and orientation are calculated and reprojected onto the video of four cameras. The videos are saved using a compression algorithm that only records the sections of the image that are moving (Straw et al. 2011). Thus, birds disappear from the video when they land and stop moving. The trajectory shown in *Figure 1* is taken from the bird labeled #2 and begins at 5.1 seconds and ends at 8.05 seconds.

individuals alone and in the presence of a competitor. Flight trajectories were parsed into a set of performance metrics based on body position and orientation. The first goal of our study was to determine if voluntary maneuvering performance is repeatable within individuals. Repeatability of maneuvering performance can arise either through a strong influence of fixed traits such as morphology and anatomy, or through other consistent influences, such as motivation. We expect that repeatable measurements will be most useful for our second goal, determining how variation in maneuverability among individuals is influenced by natural variation in morphology and muscle capacity. This also required measuring morphological traits and maximum burst performance for each individual. Our third goal was to determine how motivation state induced by the presence of a competitor influenced maneuvering performance. To address this question we compared flight trials with and without competitors.

## Results

### Maneuvering performance metrics

The first stage of analysis was estimating instantaneous velocities, accelerations, and headings from the raw tracking data (*Figure 1—figure supplement 1*). Translational velocity and acceleration were calculated by taking the first and second derivatives of an interpolation spline fit to the body position data (splev and splrep functions, Scientific Python). The velocities and accelerations were split into vertical and horizontal components. The body orientation vector was represented in spherical coordinates as azimuth and pitch angles. We took the first derivatives to obtain azimuth and pitch velocities. Because the video tracking system did not allow a measurement of body roll, we decided to use a global coordinate system instead of a body axis-centered coordinate system. In our frame of reference, pitch is a global measure defined relative to the horizontal plane. Heading was calculated as the instantaneous direction of the horizontal translation velocity, and the heading velocity was calculated as the derivative of heading.

We then used the velocity, acceleration, and orientation data to search for a series of ten stereotyped maneuvers that were independent of time and distance scales (*Figure 1b*). Five of the

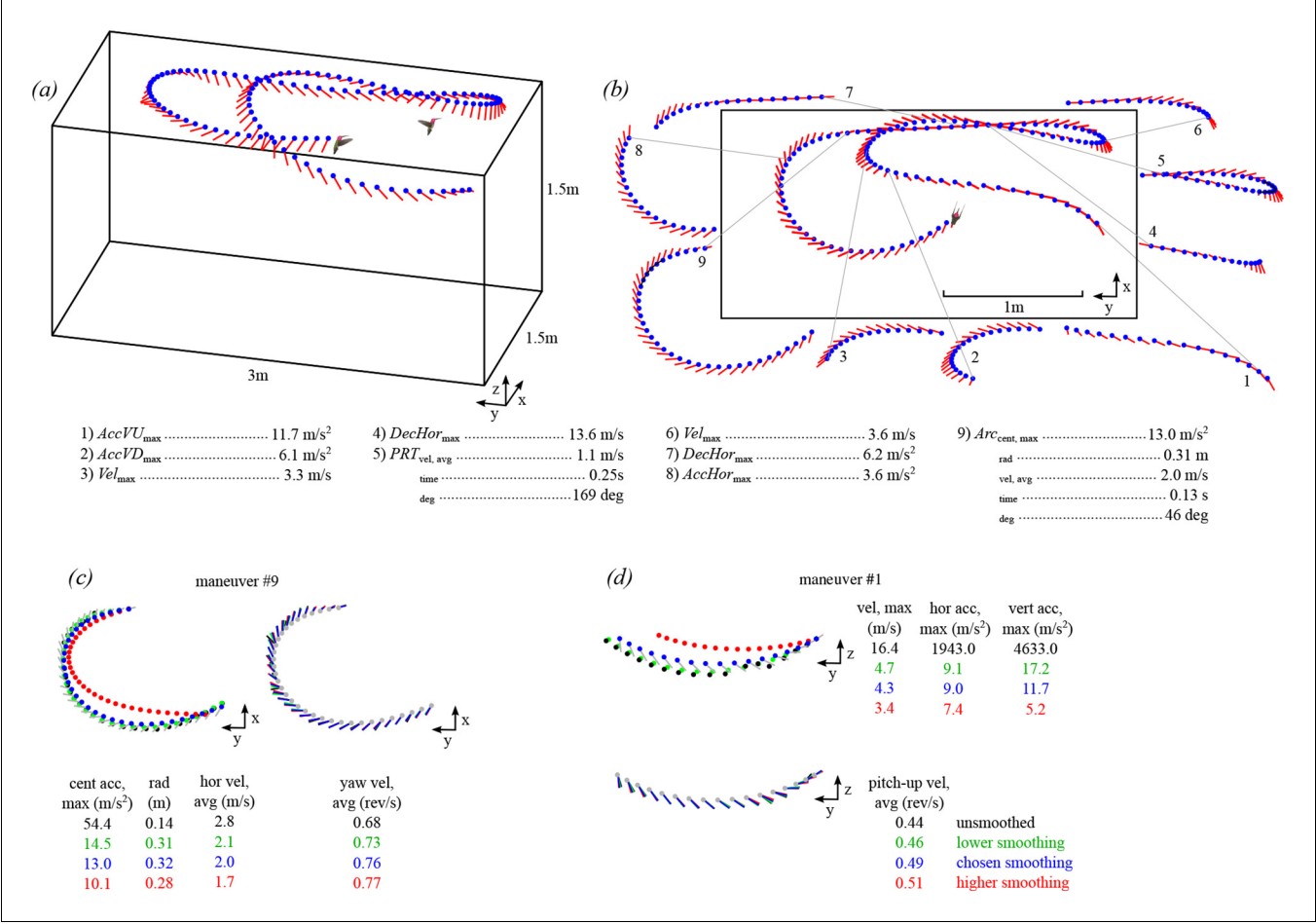

**Figure 1.** A multi-camera, automated tracking system extracted hummingbird body position (blue circle) and orientation (red line) from solo and competitive flights. The trajectory shown for one bird (a) is also shown in **Video 1** (see **Figure 1—figure supplement 1** for time series of position, velocity, and acceleration values). Stereotyped maneuvers were classified in each trajectory (b) and between one and five performance metrics were calculated from each maneuver. Maneuvers within a trajectory may be overlapping (e.g. #4,5,6). The trajectory presented in b is a top down (x-y projection) view of the trajectory shown in a. Body position and orientation were smoothed with an extended Kalman filter (c,d). The effects of four different sets of smoothing parameters are presented for an arcing turn (maneuver #9 in b) and an upward acceleration (maneuver #1 in b). Shown here are the unsmoothed position and orientation (black trace and text), the chosen levels of smoothing (blue), a lower level of smoothing (green; 0.1 x $R_{pos}$; 0.1 x $R_{ori}$), and a higher level of smoothing (red; 10 x $R_{pos}$; 10 x $R_{ori}$). The chosen smoothing parameters for body position were determined by tracking multiple dropped objects and calibrating the Z-axis acceleration to gravity. The chosen smoothing parameters for body orientation were determined by re-projecting the body axis vector onto the video. The higher and lower levels of smoothing for body position presented in this figure were both deemed too extreme, when re-projected onto the video. However, the level of smoothing for body orientation had minimal effect on the average yaw velocity.

The following figure supplement is available for figure 1:

**Figure supplement 1.** The representative trajectory from **Figure 1** and **Video 1** displayed through time.

maneuvers were sequences defined by changes in translational velocity: 1) 3D accelerations, 2) horizontal accelerations, 3) horizontal decelerations, 4) vertical upward accelerations, and 5) vertical downward accelerations. Three maneuvers were sequences defined by changes in rotation: 6) pitch-up rotations, 7) pitch-down rotations, and 8) yaw turns. Two of the maneuvers were defined as turns with translational components: 9) arcing turns and 10) pitch-roll turns. These ten maneuvers are not meant to be mutually exclusive, exhaustive, or to divide the entire filming session into a set of discrete behaviors, but are instead intended to extract simple measurements that can be used as an assay for maneuvering performance. The search criteria for the maneuvers are given in **Table 1**. Because we assume that a new maneuver must involve a change in velocity, the first search

**Table 1.** Search parameters for the ten maneuvers analyzed in the study. The definitions, units, and symbols for the 14 related performance metrics are also provided.

| Maneuver | Search parameters | Performance metric | Units | Symbol |
|---|---|---|---|---|
| 3D acceleration | Start: velocity xyz minimum<br>End: velocity xyz maximum<br>Distance xyz > 25 cm | Maximum velocity | m/s | $Vel_{max}$ |
| Horizontal acceleration | Start: velocity xy minimum<br>End: velocity xy maximum<br>Distance xy > 25 cm<br>Distance z < 10 cm | Maximum acceleration xy | m/s$^2$ | $AccHor_{max}$ |
| Horizontal deceleration | Start: velocity xy maximum<br>End: velocity xy minimum<br>Distance xy > 25 cm<br>Distance z < 10 cm | Maximum deceleration xy | m/s$^2$ | $AccDec_{max}$ |
| Vertical upwards acceleration | Start: velocity z minimum<br>End: velocity z maximum<br>Distance z > 25 cm | Maximum acceleration z | m/s$^2$ | $AccVU_{max}$ |
| Vertical downwards acceleration | Start: velocity z maximum<br>End: velocity z minimum<br>Distance z > 25 cm | Maximum acceleration z | m/s$^2$ | $AccVD_{max}$ |
| Pitch-up rotation | Start: pitch minimum<br>End: pitch maximum<br>Degrees rotated > 45 deg<br>Distance xyz < 10 cm | Average pitch velocity | rev/s | $PitchU_{vel,avg}$ |
| Pitch-down rotation | Start: pitch maximum<br>End: pitch minimum<br>Degrees rotated > 45 deg<br>Distance xyz < 10 cm | Average pitch velocity | rev/s | $PitchD_{vel,avg}$ |
| Yaw turn | Start: velocity yaw = 0 deg/s<br>End: velocity yaw = 0 deg/s<br>Degrees rotated > 90 deg<br>Pitch maximum < 75 deg<br>Distance xyz < 10 cm | Average yaw velocity | rev/s | $Yaw_{vel,avg}$ |
| Arcing turn | Start: Δ heading velocity > 0.25 rev/s<br>End Δ heading velocity < 0.25 rev/s<br>Velocity xy min > 50 cm/s<br>Distance xy > 25 cm<br>Distance z < 10 cm | Average xy velocity*<br>radius*<br>Centripetal acceleration* | m/s<br>m<br>m/s$^2$ | $Arc_{vel, avg}$<br>$Arc_{rad}$<br>$Arc_{cent, max}$ |
| Pitch roll turn | Start: velocity maximum<br>End: velocity maximum<br>Pitch maximum > 75 deg<br>Distance xy before velocity<br>Min > 12.5 cm<br>Distance xy after velocity<br>Min < 12.5 cm<br>Distance z < 10 cm | time[†]<br>degrees turned[†] | s<br>deg | $PRT_{time}$<br>$PRT_{deg}$ |

*for a 25 cm segment centered at the sharpest point of the turn
[†]for a 25 cm segment centered at the minimum velocity xyz

parameter was to find sequences bounded by velocity maxima and minima, or vice versa. We next describe the additional search parameters and the performance metrics used to quantify each maneuver.

The five translational maneuvers were defined using velocity minima and maxima, and only sequences with at least 25 cm of travel were analyzed. The 3D acceleration maneuvers started from a velocity minimum and ended with a velocity maximum. The performance metric calculated for these maneuvers was the maximum translational velocity ($Vel_{max}$). The horizontal acceleration maneuvers were bounded by horizontal velocity minima and maxima, and were constrained to no more than 10 cm of vertical distance traveled. The performance metric calculated for these maneuvers was the maximum horizontal acceleration ($AccHor_{max}$). The horizontal deceleration maneuvers and the corresponding performance metric, maximum horizontal deceleration ($DecHor_{max}$), were

bounded by horizontal velocity maxima and minima. The vertical upward acceleration and vertical downward acceleration maneuvers were bounded by vertical velocity minima and maxima. The performance metrics calculated from these maneuvers were, respectively, maximum upward ($AccVU_{max}$) and maximum downward ($AccVD_{max}$) accelerations. All translational accelerations and decelerations were expressed as positive values, so that higher values represent a higher level of performance.

We defined three rotational maneuvers: pitch-up rotations, pitch-down rotations, and yaw turns. These sequences were bounded by the zero-crossings of the azimuthal and pitch velocities. In contrast to translational maneuvers, which were defined by the maxima and the minima of the velocities, the rotational maneuvers begin and end with changes in rotational velocity direction. Thus, the performance metrics calculated from these rotational maneuvers were the average rotational velocities over the whole maneuver instead of maximum accelerations or decelerations. An additional constraint common to all three rotational maneuvers is that the linear distance traveled was less than 10 cm. We chose 10 cm as a general cutoff here and elsewhere because this value is close to the body length of a bird and the wing span at mid-downstroke, thus providing a good threshold for distinguishing translational motion.

The pitch-up and pitch-down maneuvers were defined as having continuous pitch velocity in the upward or downward direction, respectively. Only maneuvers with a total pitch rotation greater than 45° were analyzed. From these maneuvers we calculated either the average pitch-up ($PitchU_{vel, avg}$) or pitch-down ($PitchD_{vel, avg}$) velocity as performance metrics. Defining yaw turns is challenging because hummingbirds fly with an upright body posture. When the body posture is near vertical, azimuthal rotation is implemented by rolling about the body axis, but when the body posture is near horizontal, azimuthal rotation is implemented by yawing the body axis. We therefore define yaw turns as azimuthal changes in direction when the body pitch angle is below 75°. An additional constraint specific to yaw turns was a requirement for at least 90° change in azimuth. From these trajectories we measured the average yaw velocity ($Yaw_{vel, avg}$) as the performance metric.

In addition to five translational and three rotational maneuvers, we also considered two maneuvers that are complex turns with translational components. Arcing turn maneuvers were defined as sequences with a heading velocity > 90°/sec, a minimum total translational velocity > 0.5 m/s, a total distance traveled > 25 cm, and a vertical distance traveled < 10 cm. These search parameters reliably extract arcing turns that occur in the horizontal plane. To compare arcing turns of different shapes and scales we clipped the trajectories to a length of 25 cm centered at the sharpest point of the turn. From the clipped trajectory we analyzed three performance metrics, average velocity ($Arc_{vel, avg}$), radius ($Arc_{rad}$), and the maximum centripetal acceleration ($Arc_{cent, max}$). The latter two were calculated using the following equations:

$$Arc_{rad} = \frac{Arc_{distance\ traveled}}{\Delta Heading_{rad}}$$

$$Arc_{cent,max} = \frac{Arc_{vel,avg}^2}{Arc_{rad}}$$

Pitch-roll turn maneuvers have been described in hummingbirds and are characterized by the following sequence: a) deceleration, b) increase in pitch to near vertical, c) azimuthal rotation by rolling the body, and d) acceleration in a new direction (*Clark, 2011*). These maneuvers were identified by searching for sequences of deceleration followed by acceleration with a maximum pitch > 75°. Just as we did for the yaw turns, we assume that above a pitch angle of 75°, the rotation is primarily dominated by a body axis roll, even if there may be a slight yawing component. For this reason, we maintain the established 'pitch-roll' terminology to describe these types of turns. These sequences were clipped to a linear distance of 25 cm centered on the point of the lowest translational velocity. Only clipped sequences in which the total vertical displacement was less than 10 cm were analyzed. The performance metrics for pitch-roll turns were the time taken ($PRT_{time}$) and the degrees turned ($PRT_{deg}$).

Arcing turns and pitch-roll turns are two different mechanisms for generating a change in heading with no overlap in our data set by definition (*Table 1*). We analyzed how morphology, burst capacity, and competitor presence influenced the relative use of these two turns. The pitch-roll percent (PRT

**Table 2.** Wing morphology and load lifting performance of male Anna's hummingbirds (n = 20 individuals).

| Trait | Mean | Range |
|---|---|---|
| Wing length | 50.97 mm | [45.76, 55.45] |
| Wing area | 1355 mm$^2$ | [1051, 1653] |
| Wing aspect ratio | 7.73 | [7.13, 8.46] |
| Body mass | 4.64 g | [4.09, 5.61] |
| Mass of weights lifted | 5.93 g | [4.00, 7.24] |

%) was defined as the number of pitch-roll turns divided by total the number of arcing and pitch-roll turns extracted from each trial.

## Descriptive statistics

Descriptive statistics for morphology and load lifting are provided in *Table 2*. A large sample of values was obtained for each maneuvering performance metric (*Table 3*). *Figure 2* shows the distributions of trial means for all performance metrics.

## Repeatability of performance

All performance metrics based on total and horizontal linear accelerations and complex turns were highly repeatable, with >80% of the variation in these metrics attributable to differences among individuals (*Figure 3*). The rotational performance metrics and the percent of turns that were pitch-roll turns were moderately repeatable, with 40–70% of the variation in these metrics attributable to

**Table 3.** Descriptive statistics and sample sizes for maneuvering performance. Grand mean values were calculated by first taking the mean of each bird's trial averages (i.e., the bird means), and then taking the mean of the bird means (*n* = 20 birds in 20 solo trials and 16 paired competition trials).

| Maneuverability | Performance metric | # Trajectories | Grand mean | [Range of means] |
|---|---|---|---|---|
| Linear accelerations | $Vel_{max}$ | 71,007 | 2.22 m/s | [1.20, 2.94] |
| | $AccHor_{max}$ | 47,287 | 6.30 m/s$^2$ | [2.96, 8.83] |
| | $DecHor_{max}$ | 51,245 | 6.67 m/s$^2$ | [9.03, 3.45] |
| | $AccVU_{max}$ | 6,935 | 3.78 m/s$^2$ | [2.98, 4.67] |
| | $AccVD_{max}$ | 9,284 | 3.58 m/s$^2$ | [4.69, 2.68] |
| | | | | |
| Rotational velocities | $PitchU_{vel,\ avg}$ | 6,085 | 1.13 rev/s | [0.91, 1.34] |
| | $PitchD_{vel,\ avg}$ | 14,807 | 1.00 rev/s | [1.19, 0.78] |
| | $Yaw_{vel,\ avg}$ | 12,660 | 1.52 rev/s | [1.32, 1.75] |
| | | | | |
| Complex turns | | | | |
| Pitch-roll | $PRT_{deg}$ | 17,133 | 133.3 ° | [34.9, 162.7] |
| | $PRT_{time}$ | 17,133 | 0.47 s | [0.38, 0.60] |
| | | | | |
| Arcing | $Arc_{rad}$ | 6.945 | 0.48 m | [0.14, 0.70] |
| | $Arc_{vel,\ avg}$ | 6,945 | 1.57 m/s | [0.80, 2.26] |
| | $Arc_{cent,\ max}$ | 6,945 | 6.59 m/s$^2$ | [3.42, 10.80] |
| | | | | |
| Use of turns | $PRT\%$ | 24,078 | 0.69 | [0.39, 0.87] |

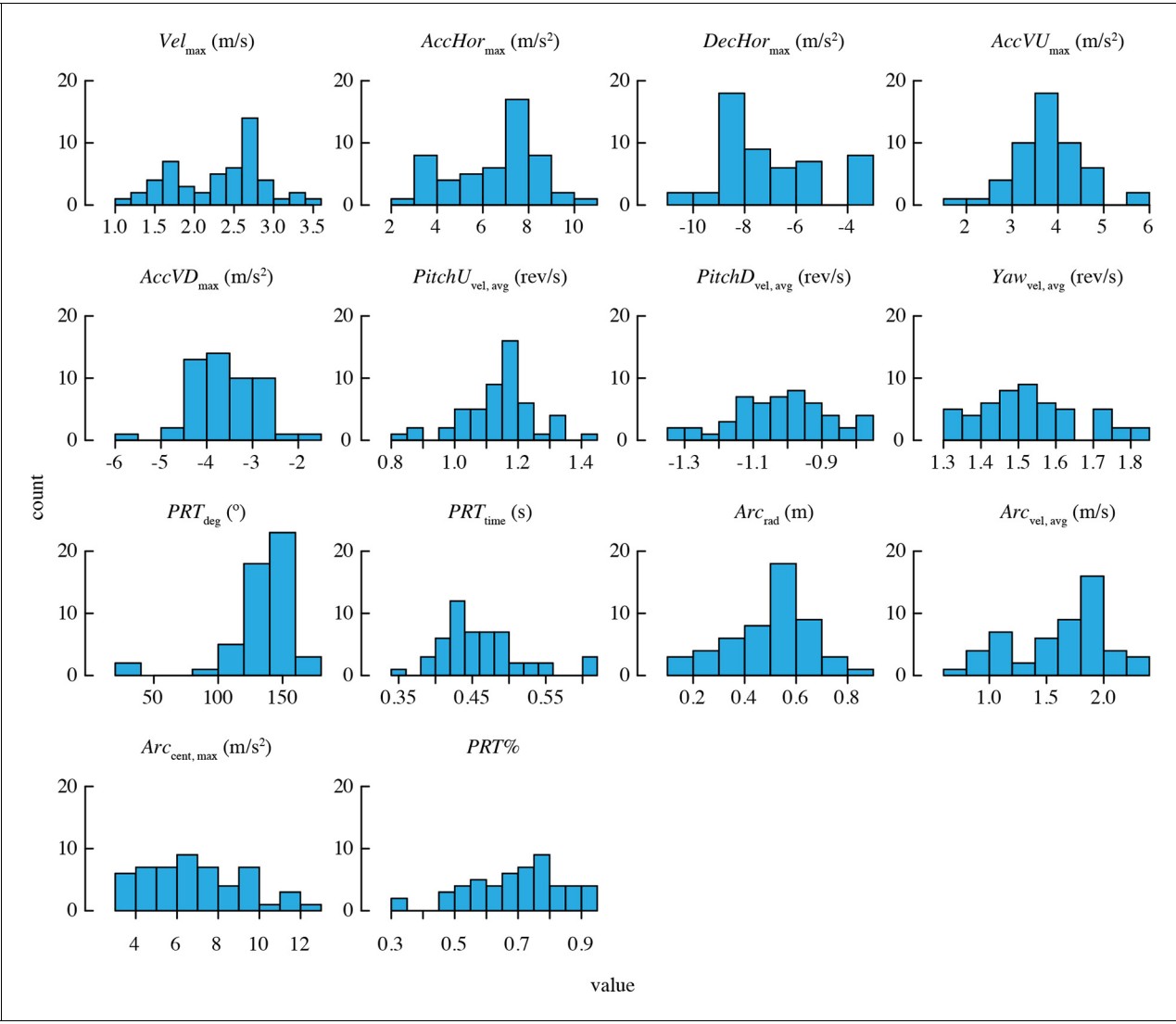

**Figure 2.** Distributions of mean performance metric values for n = 52 bird-trial combinations. Only $PRT_{deg}$ has statistically significant outliers. See *Figure 2—figure supplement 1* for distributions of residuals from the best-fit model in each case. Note that two statistical outliers were omitted from the analysis of $PRT_{deg}$.

The following figure supplement is available for figure 2:

**Figure supplement 1.** Distributions of residuals from the best-fit model for each performance metric.

among-individual differences. The vertical accelerations were not repeatable, as the 95% confidence intervals for repeatability of these metrics overlapped zero.

## Maneuvering in relation to burst muscle capacity

The best-supported models for each maneuvering performance metric are given in *Table 4*. Burst muscle capacity was an important predictor for most of the maneuvering performance metrics. Birds that lifted more weight (accounting for their wing morphology) tended to accelerate and decelerate faster, and they tended to perform maneuvers with higher velocity (*Figure 4*). However, burst muscle capacity was not an important determinant of vertical acceleration and deceleration, as candidate models including burst performance as a predictor were not supported. Birds that lifted more weight also executed pitch-up and pitch-down maneuvers with higher rotational velocities. Burst capacity was not a strong determinant of yaw performance. Although yaw velocity was somewhat

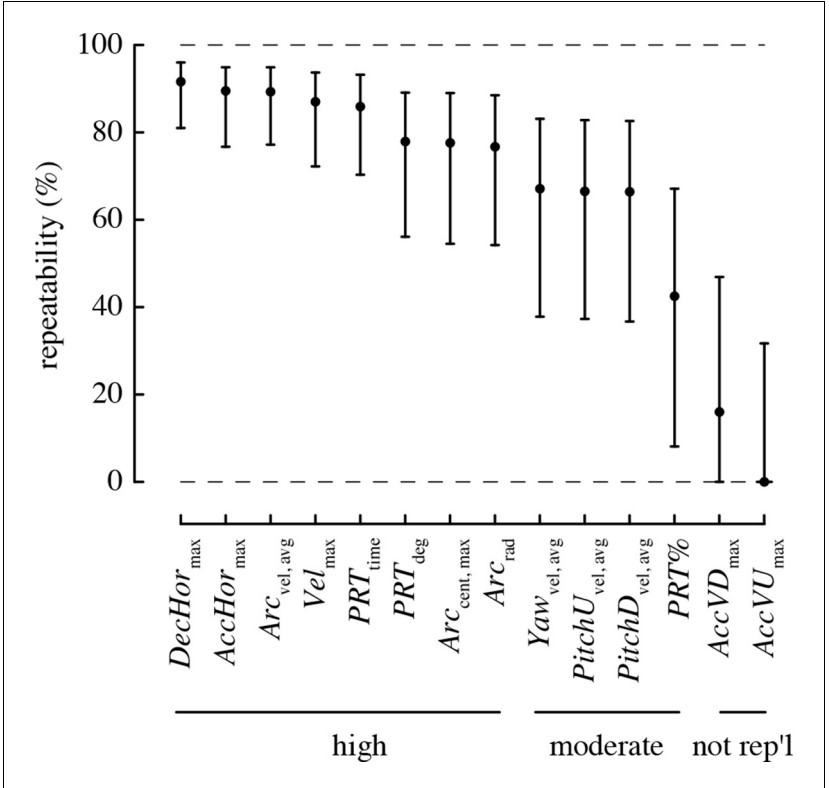

**Figure 3.** Most maneuvering performance metrics are highly repeatable. Values > 70% are considered to have high repeatability, 40–70% moderate repeatability, and < 40% low repeatability. A metric is considered not repeatable if its 95% confidence intervals overlap zero.

positively related to burst capacity (**Figure 4**), candidate models of yaw velocity that included burst as a predictor were not well supported.

Burst muscle capacity was also associated with some, but not all maneuvering performance metrics related to complex turns. Birds that lifted more weight for their wing morphology tended to execute faster, larger radius arcing turns (**Figure 4**). However, the centripetal acceleration of arcing turns was not associated with burst capacity. Hummingbirds with higher load lifting capacity executed pitch-roll turns in less time. Burst capacity was not a strong determinant of heading change during pitch-roll turns. Lastly, birds with higher burst muscle capacity used pitch-roll turns for proportionately more of their heading changes.

## Maneuvering in relation to morphology

Wing morphology, specifically the aspect ratio, was an important predictor for two performance metrics: centripetal acceleration and the percent of direction changes that were pitch-roll turns (**Figure 5**). Hummingbirds with long, narrow wings tended to perform arcing turns with higher centripetal accelerations, relative to birds with short, wide wings. Birds with higher aspect ratio wings also used proportionately more arcing turns than birds with low aspect ratio wings.

Body mass was included in candidate models 1–7 because we had anticipated that body mass would have a strong influence on variation in maneuvering performance. However, for every performance metric in **Table 4**, the coefficient estimate for body mass had confidence intervals that broadly overlapped zero.

## Effect of competitor on maneuvering performance

We did not detect a substantial effect of competitor presence on many of the performance metrics (**Table 4**). Two metrics, horizontal acceleration and deceleration, were affected, but in the direction opposite to what we predicted. Specifically, birds performed maneuvers with lower acceleration (−

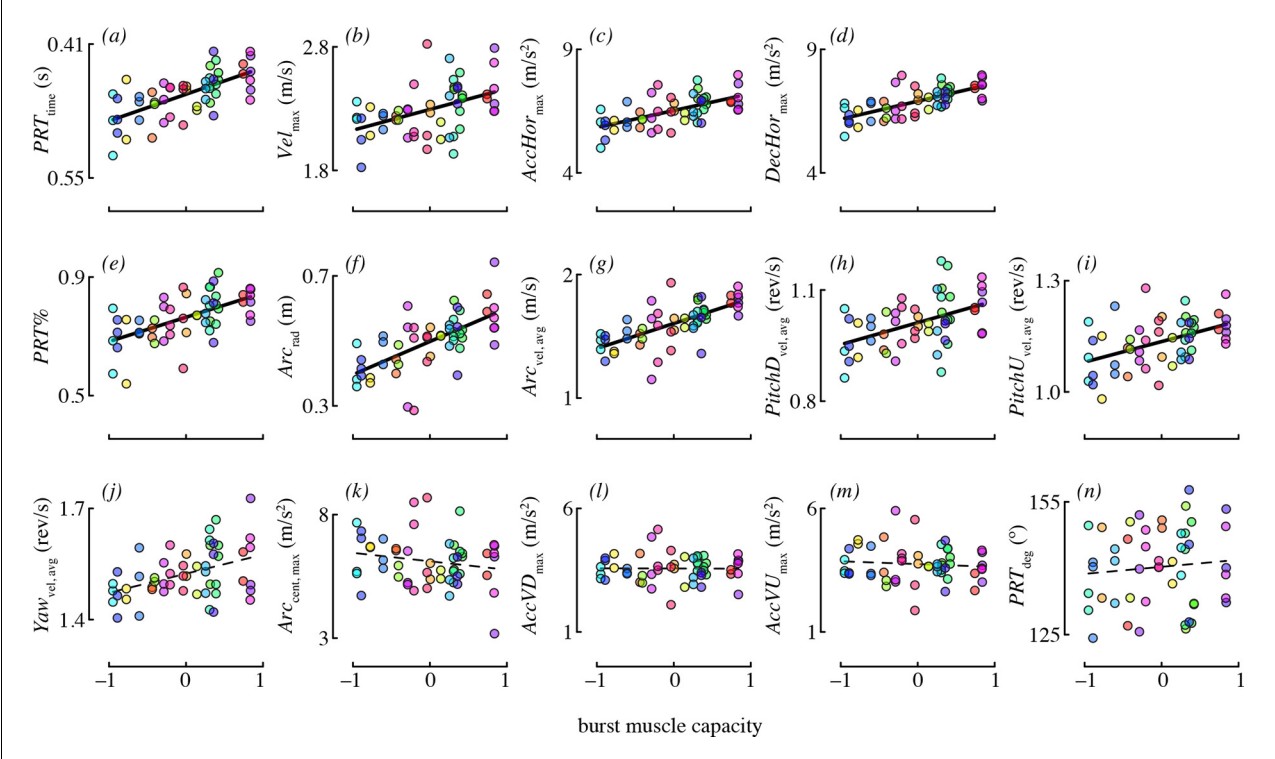

**Figure 4.** Burst muscle capacity was associated with most maneuvering performance metrics. Each panel shows partial residuals for a performance metric (y-axis) in relation to burst muscle capacity (x-axis) for the most supported candidate model with burst capacity as a predictor. Partial residual values (y-axis) account for the other fixed effects in that model. Lines show model predictions assuming the median value of continuous predictors, and averaging across experiments and levels of competitor presence. Prediction lines are dashed for metrics where burst performance was not present in any of the supported models. Color is used to denote data points from the same bird (online version only).

0.46 m/s$^2$ difference on average) and lower deceleration (–0.47 m/s$^2$) in the presence of a competitor, relative to solo flight (*Figure 6a,b*). One metric, pitch-down velocity (*Figure 6c*), did increase during competition as predicted (0.06 rev/s difference on average). We had no prediction for how competition would influence the relative use of pitch-roll and arcing turns, but found that birds used proportionately more arcing turns in the presence of a competitor (*Figure 6d*). Specifically, 35% of direction changes were arcing turns on average (and 65% pitch-roll) when a competitor was present, whereas during solo flight, only 23% of direction changes were arcing turns (and 77% pitch-roll) on average.

## Discussion

We collected a large number of free flight measurements for each of 20 individual hummingbirds to examine the biomechanical determinants of maneuverability. Other studies have measured elements of maneuvering performance of hummingbirds in the field (*Clark, 2009*; *Sholtis et al., 2015*) and documented the maximum velocities, accelerations, and rotations obtained during specific maneuvers. Our values for velocity and acceleration are considerably lower than either of the field studies, likely because of cage size. However, the benefit of using a flight chamber is that it allowed us to evaluate the relative contributions of different factors to the performance we observed. We found that hummingbirds maneuvered with highly repeatable performance (*Figure 3*). Maximum weight lifted during load lifting trials predicted most of the performance metrics that we measured, independent of a bird's wing size and shape, such that birds with higher burst muscle capacity flew faster, had higher horizontal accelerations, faster rotations, and higher performance during complex turns (*Figure 4*). Aspect ratio predicted only two performance metrics, such that birds with higher aspect ratio wings performed turns with higher centripetal acceleration and a greater percentage of

**Table 4.** Maneuvering performance in relation to burst performance, wing morphology, and competitor presence (n = 20 birds in 20 solo trials and 16 paired competition trials). Standardized beta coefficients and $R^2_{GLMM(m)}$ values are reported for either the best-fit model, or, if there was support for more than one model, the average of supported models. The standardized beta coefficient is a measure of effect size that can be compared among predictors in the same model. Relative importance is a measure of the weight of evidence in favor of a predictor on a scale from 0–1, and is reported for burst capacity and wing morphology variables as these alone were subject to model selection. Marginal $R^2_{GLMM(m)}$ provides a measure of the combined explanatory power of fixed effects of interest (competitor presence, burst muscle capacity, and wing morphology effects combined). Details of all candidate models are provided in **Supplementary file 1**.

| Model | Support for | Fixed effects | Std beta coef [95% CI] | Relative importance | $R^2_{GLMM(m)}$ Burst + morphology + competitor |
|---|---|---|---|---|---|
| $Vel_{max}$ | burst | competitor presence<br>mass<br>burst<br>wing length<br>wing aspect ratio<br>experiment (CA1)<br>experiment (CA2)<br>days post-capture | –0.04 [–0.18, 0.10]<br>0.10 [–0.01, 0.22]<br>0.09 [0.00, 0.18]<br>–0.08 [–0.22, 0.06]<br>0.10 [–0.07, 0.28]<br>1.01 [0.59, 1.42]<br>1.06 [0.68, 1.43]<br>–0.07 [–0.24, 0.11] | –<br>–<br>1.00<br>0.25<br>0.26<br>–<br>–<br>– | 0.28 |
| $AccHor_{max}$ | burst + competition | competitor presence<br>mass<br>burst<br>experiment(CA1)<br>experiment(CA2)<br>days post-capture | –0.46 [–0.82, –0.11]<br>0.20 [–0.28, 0.69]<br>0.39 [0.00, 0.77]<br>4.01 [2.46, 5.56]<br>3.68 [2.72, 4.64]<br>–0.39 [–1.09, 0.32] | –<br>–<br>1.00<br>–<br>–<br>– | 0.18 |
| $DecHor_{max}$ | burst + competition | competitor presence<br>mass<br>burst<br>experiment(CA1)<br>experiment(CA2)<br>days post-capture | –0.47 [–0.78, –0.16]<br>0.31 [–0.13, 0.74]<br>0.41 [0.06, 0.76]<br>3.86 [2.47, 5.25]<br>3.64 [2.76, 4.51]<br>–0.24 [–0.88, 0.39] | –<br>–<br>1.00<br>–<br>–<br>– | 0.19 |
| $AccVU_{max}$ | intercept-only | NA | NA | NA | 0 (NA) |
| $AccVD_{max}$ | intercept-only | NA | NA | NA | 0 (NA) |
| $PitchU_{vel, avg}$ | burst | competitor presence<br>mass<br>burst<br>experiment(CA1)<br>experiment(CA2) | 0.02 [–0.02, 0.06]<br>0.00 [–0.04, 0.04]<br>0.03 [–0.01, 0.07]<br>0.14 [0.06, 0.23]<br>0.13 [0.03, 0.23] | –<br>–<br>1.00<br>–<br>– | 0.10 |
| $PitchD_{vel, avg}$ | competition + burst | competitor presence<br>mass<br>burst<br>wing length<br>wing aspect ratio<br>experiment(CA1)<br>experiment(CA2) | 0.06 [0.01, 0.10]<br>0.01 [–0.04, 0.05]<br>0.03 [–0.01, 0.08]<br>0.04 [–0.03, 0.12]<br>–0.04 [–0.13, 0.05]<br>0.19 [0.03, 0.34]<br>0.22 [0.03, 0.41] | –<br>–<br>0.66<br>0.37<br>0.28<br>–<br>– | 0.18 |
| $Yaw_{vel, avg}$ | intercept-only | NA | NA | NA | 0 (NA) |
| $PRT_{deg}$ | intercept-only | NA | NA | NA | 0 (NA) |
| $PRT_{time}$ | burst | competitor presence<br>mass<br>burst<br>wing length<br>wing aspect ratio<br>experiment(CA1)<br>experiment(CA2) | 0.00 [–0.01, 0.01]<br>–0.01 [–0.03, 0.00]<br>–0.02 [–0.03, 0.00]<br>–0.01 [–0.03, 0.01]<br>0.01 [–0.01, 0.04]<br>–0.08 [–0.12, –0.03]<br>–0.11 [–0.16, –0.05] | –<br>–<br>1.00<br>0.21<br>0.23<br>–<br>– | 0.29 |
| $Arc_{rad}$ | burst | competitor presence<br>mass<br>burst<br>wing aspect ratio<br>experiment(CA1)<br>experiment(CA2) | –0.02 [–0.07, 0.03]<br>0.01 [–0.03, 0.06]<br>0.06 [0.01, 0.10]<br>–0.06 [–0.15, 0.03]<br>0.25 [0.12, 0.37]<br>0.29 [0.06, 0.52] | –<br>–<br>1.00<br>0.44<br>–<br>– | 0.22 |

*Table 4 continued on next page*

*Table 4 continued*

| Model | Support for | Fixed effects | Std beta coef [95% CI] | Relative importance | $R^2_{GLMM(m)}$ Burst + morphology + competitor |
|---|---|---|---|---|---|
| $Arc_{vel,\ avg}$ | burst | competitor presence | –0.01 [–0.09, 0.08] | – | 0.18 |
| | | mass | 0.03 [–0.06, 0.12] | – | |
| | | burst | 0.11 [0.04, 0.19] | 1.00 | |
| | | experiment(CA1) | 0.89 [0.59, 1.19] | – | |
| | | experiment(CA2) | 0.74 [0.56, 0.92] | – | |
| | | days post-capture | –0.06 [–0.20, 0.08] | – | |
| $Acc_{cent,\ max}$ | wing shape | competitor presence | 0.29 [–0.37, 0.94] | – | 0.36 |
| | | mass | –0.20 [–0.74, 0.34] | – | |
| | | wing aspect ratio | 1.09 [0.19, 1.99] | 1.00 | |
| | | experiment(CA1) | 5.93 [4.02, 7.84] | – | |
| | | experiment(CA2) | 0.85 [–1.59, 3.28] | – | |
| | | days post-capture | –1.76 [–2.62, –0.90] | – | |
| PRT% | wing shape + competition + burst + wing size | competitor presence | –0.14 [–0.19, –0.09] | – | 0.27 |
| | | mass | 0.00 [–0.04, 0.05] | – | |
| | | burst | 0.04 [0.00, 0.09] | 1.00 | |
| | | wing length | –0.06 [–0.13, 0.01] | 0.61 | |
| | | wing aspect ratio | –0.16 [–0.24, –0.07] | 1.00 | |
| | | experiment(CA1) | 0.17 [–0.03, 0.36] | – | |
| | | experiment(CA2) | 0.44 [0.19, 0.69] | – | |

arcing turns (*Figure 5*). When flying in the presence of a competitor, hummingbirds used faster pitch velocities, although they used slower horizontal accelerations and decelerations. During competition trials birds also increased the proportion of arcing turns used (*Figure 6*). Collectively, these results suggest that burst muscle capacity is a much more important predictor of flight maneuverability than within-species variation in body mass, wing morphology, and competition with conspecifics.

Why were body mass and wing size not associated with maneuvering performance? Wing morphology has well-known physical affects on flight performance: aspect ratio predicts aerodynamic efficiency, wing area is directly proportional to aerodynamic force, and wing length is a strong predictor of wingbeat frequency. All of these morphological traits, along with body mass, could affect maneuverability in flight, either individually or in combination. For example, wing loading (the ratio of body mass to wing area or to area swept by the wings) was initially thought to be a key predictor of hummingbird flight performance and behavioral ecology (*Feinsinger and Chaplin, 1975*; *Feinsinger, 1976*; *Feinsinger and Colwell, 1978*; *Feinsinger et al., 1979*). However, in our analysis the hypothesis that wing size and body mass together determine maneuvering performance was not supported for any performance metric (see *Supplementary file 1*). We found it especially surprising that only wing shape (and not wing size) predicted maneuvering performance. It is possible that other morphological traits may determine maneuvering performance, or that subtle relationships may have gone undetected, because our analysis was limited to 20 individuals of a single species. It would be informative to expand this analysis to other species with potentially greater within-species variation in wing morphology, and to assess maneuverability across different hummingbird species with divergent morphologies.

Almost all of the performance metrics were highly repeatable, which indicates a potential role for intrinsic influences of wing morphology in determining maneuverability. However, aspect ratio was the only morphological parameter that predicted performance, and only for a limited set of maneuvers. Aspect ratio is a key determinant in wing efficiency for fixed wings, such as during gliding (*Pennycuick, 1983*), and it has recently been demonstrated that higher aspect ratio wings correspond to higher power factors in the revolving wings of hummingbirds (*Kruyt et al., 2014*). We found that aspect ratio had a strong effect on the few performance metrics that it predicted, but did not affect most features of maneuvering performance. This suggests a limited role for aerodynamic efficiency in many features of maneuvering.

Burst muscle capacity predicted most of the performance metrics we considered, independently of any association with wing size or shape. Load lifting is measured as a transient escape maneuver that is likely anaerobic and performed inefficiently. All hummingbirds reach maximum load lifting

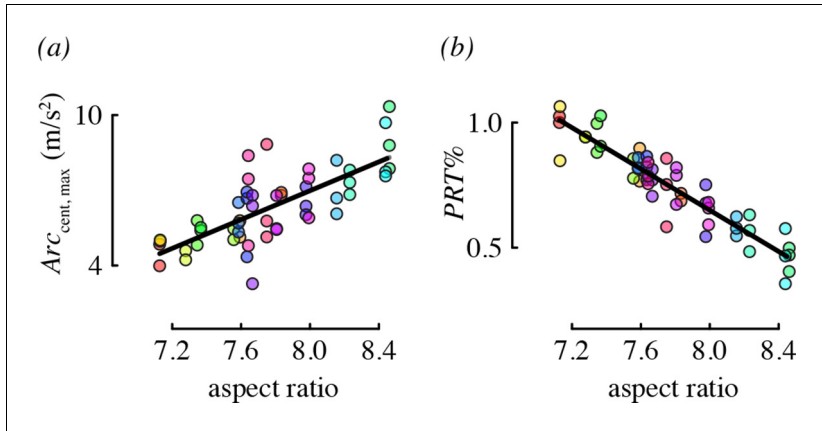

**Figure 5.** Aspect ratio was associated with two maneuvering performance metrics. Each panel shows partial residual performance (y-axis) in relation to wing aspect ratio (x-axis) from a best-fit model that identified aspect ratio as an important predictor. Note that the partial residuals for PRT% in (**b**) go above 1 because PRT% was modeled as a normally-distributed (Gaussian) variable. All other features as in *Figure 4*.

performance at a geometric limit set by the amplitude of the wings: wing stroke amplitude cannot extend much past 180° without the two wings interfering with each other physically and aerodynamically (*Chai and Dudley, 1995*; *Chai et al., 1997*; *Chai and Millard, 1997*; *Altshuler and Dudley, 2003*). Maximum load lifting also elicits a substantial increase in wingbeat frequency as a constant fraction of baseline wingbeat frequency (*Altshuler and Dudley, 2003*). Thus, maximum load lifting performance involves brief increases in muscle strain and muscle velocity to physically imposed limits. The capacity to increase muscle strain and velocity has previously been shown to influence foraging behavior and competitive ability (*Altshuler, 2006*). The results of the current study demonstrate that it also underlies multiple features of maneuvering performance.

The two performance metrics that were not repeatable are vertical accelerations and decelerations, which were expected to be important based on previous observations of hummingbird competitive interactions (*Altshuler, 2006*) and mating displays (*Clark, 2009*). Moreover, vertical performance was not well predicted by morphology, burst capacity, or competitor presence in this study. The dimensions of our experimental chamber likely influenced our observations of vertical performance. Hummingbirds in captivity tend to fly near the top of their cages, and the vertical dimension of the chamber (1.5 m) may have limited vertical movement.

Male hummingbirds are extremely aggressive towards conspecifics (*Kodric-Brown and Brown, 1978*; *Carpenter et al., 1983*) and other species of hummingbirds (*Stiles and Wolf, 1970*; *Wolf et al., 1976*). The most territorial species will vigorously defend territories (*Carpenter et al., 1983*) and lekking sites (*Rico-Guevara and Araya-Salas, 2015*). In staged competition studies, paired hummingbirds will also establish and defend territories (*Tiebout, 1993*). We originally intended to use competition to elicit high levels of flight activity and maneuvering performance in territorial male Anna's hummingbirds (*Stiles, 1982*). However, we found that competitor presence affected only a small number of the maneuvering performance metrics that we measured. Pitch-down velocity increased with competition whereas horizontal acceleration and deceleration actually decreased. We do not know why these three metrics (in addition to *PRT%*; see below) were strongly affected by competition or why they were affected in the directions observed. However, there are several possible causes for why competitor presence did not affect the other metrics: 1) we were unable to elicit a high level of competition or territoriality; 2) the birds may have worked out dominance without the aggressive interactions normally seen outdoors; and/or 3) the interactions required to establish dominance may have been very brief (*Maynard Smith, 1974*) such that they comprised only a minuscule sample of the maneuvers we analyzed. This experiment was not designed to study the effects of maneuvering performance on competitive success, although this represents an important topic for future investigation. Laboratory performance tests do not always reflect field behavior (*Irschick, 2003*) and outdoor studies of maneuvering performance will be

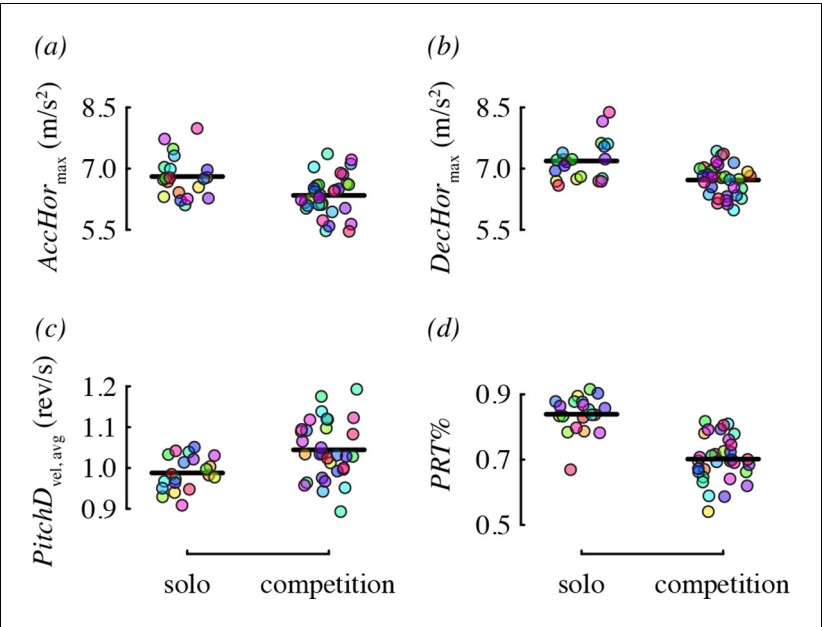

**Figure 6.** Competitor presence was associated with four maneuvering performance metrics. Each panel shows residual performance (y-axis) in relation to competitor presence from a best-fit model where competitor presence had a detected effect. All other features as in *Figure 4*.

important for understanding the role of maneuverability in competitive interactions. Recent advances in video tracking (*Theriault et al., 2014*; *Shelton et al., 2014*) should make it possible to track individuals for multiple measurements.

The most substantial result of competitor presence was the increase in the use of arcing over pitch-roll turns. These two types of turns represent different strategies for changing direction that differ in duration and amount of heading change. Arcing turns require less time but are used for smaller heading changes, whereas pitch-roll turns are longer but can be used to change heading by 180° (*Figure 7*). Given that hummingbird agonistic interactions can involve direct contact and stabbing with bills (*Tiebout, 1993*; *Clark and Russell, 2012*; *Rico-Guevara and Araya-Salas, 2015*), slow turns in place could make a bird more vulnerable during competition.

The relative use of arcing and pitch-roll turns was the only metric in our study that was influenced by all of morphology, burst muscle capacity, and competitor presence. The minimum radius of an arcing turn is limited by the maximum centripetal acceleration that a bird can generate while maintaining lift. The speed of a pitch-roll turn is limited by the ability to decelerate and then accelerate. Birds with higher wing aspect ratio may have preferred arcing turns because they were able to generate higher centripetal accelerations. Birds with higher burst muscle capacity may have favored pitch-roll turns because they had higher accelerating and decelerating performance. These observations suggest the hypothesis that high aspect ratio and high burst capacity enhance maneuverability. This hypothesis could be evaluated by comparing hummingbird species that differ in wing shape, foraging strategy, and burst capacity (*Altshuler et al., 2004b*; *2010a*; *Altshuler, 2006*; *Kruyt et al., 2014*).

By constraining hummingbirds to fly in a large chamber we were able to track and measure a large sample of maneuvers attributed to individuals with known morphological traits and burst performance. A major contribution of our study is the development of an assay of free flight maneuvering performance based on large numbers of stereotyped movements. Using this method, we identify several performance metrics that were highly repeatable across trials for individual hummingbirds, strongly correlated with individual morphological and physiological characteristics, and largely uninfluenced by the added motivation of a conspecific competitor. This approach to measuring maneuverability will be useful for future studies comparing maneuvering performance across

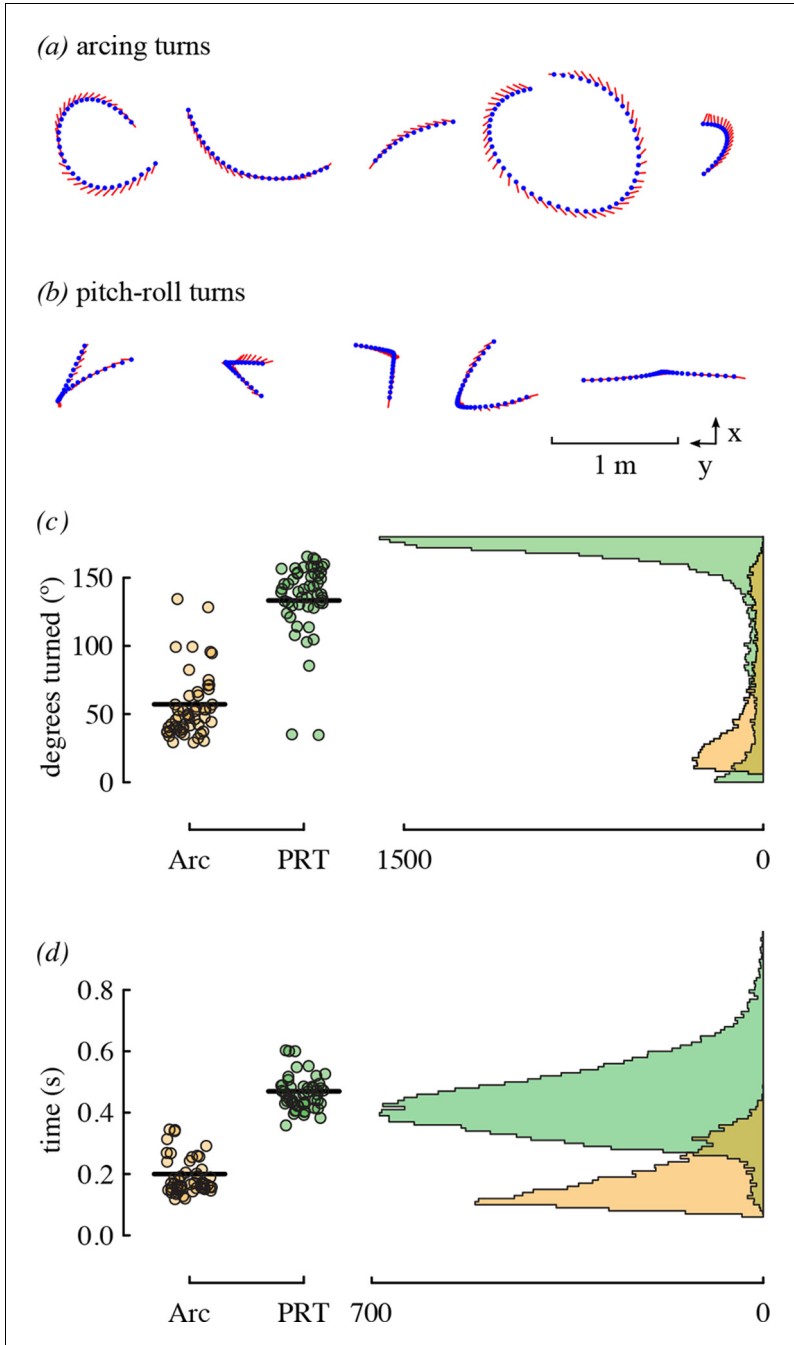

**Figure 7.** Arcing and pitch-roll turns are two classes of complex maneuver that differ in turn magnitude and duration. Representative examples of arcing (a) and pitch-roll (b) turns are depicted from the above perspective. Arcing turns (Arc; orange) and pitch-roll turns (PRT; green) differed in the degrees turned (c) and elapsed time (d). Circles represent bird-trial means (n = 52) with grand means indicated with black lines. Histograms for the pooled dataset of all maneuvers are given on the right. The outliers for degrees turned in pitch-roll turns were included when calculating the grand means but not in the model analyses (*Table 4*).

different experimental manipulations, geographic ranges, or ecological, morphological, and phylogenetic groups.

## Materials and methods

### Animals and experimental trials

We captured and filmed 20 adult male Anna's hummingbirds (*Calypte anna*) at the University of California, Riverside (eight birds in July-October 2009; four birds in January-March 2010) and the University of British Columbia (eight birds in December 2013-April 2014). The hummingbirds were housed in individual cages and fed ad libitum with a solution of artificial nectar (Nektar-Plus, Nekton, Pforzheim, Germany) and sucrose. The flight arenas were large rectangular cages (3 x 1.5 x 1.5 m) built with an aluminum frame and had either garden mesh (California) or clear acrylic (British Columbia) side panels. The cages contained multiple perches and a single feeder hung from the roof of the cage.

Before the first trial, each bird was allowed to acclimate to the flight arena and learn where the perches and the feeder were located. The trials began once the birds were actively exploring the cage and consistently visiting the feeder and both perches. At this point, we recorded high-speed video of a two-hour solo trial for each bird. Following solo trials (between 0–23 days later), birds were paired and filmed for another two hours in competition trials. One bird in each pair was marked with a small square of retro-reflective tape placed between the shoulder blades for identification. The birds filmed in British Columbia had one competition trial and the birds in California had two competition trials. In the latter case, the second trial consisted of previously unknown opponents that were chosen randomly from the remaining pool. The competition trials involved chases, displacements, and aerial displays, but very little contact. Regardless, we monitored the competition trials to ensure that no birds were harmed or excluded from the feeder.

Following each round of solo and competition trials, we performed asymptotic load lifting experiments using the techniques described in Chai et al. (*Chai et al., 1997*), and subsequently used in other studies estimating maximum burst power output (*Chai and Millard, 1997*; *Altshuler et al., 2004a*; *2010b*; *Altshuler, 2006*). Here, we use the mass of maximum number of beads lifted by each individual as a measure of burst performance. Immediately following load lifting, we weighed the birds and photographed both wings in an outstretched position against white paper with a reference scale (*Chai and Dudley, 1995*). We oriented the wing image and divided it into pixel wide strips representing the wing chords at each value of wing radius. Values for aspect ratio, wing area, and wing length were then calculated based on equations in Ellington (*Ellington, 1984*).

We considered wing area and wing length as two potential measures of wing size, but these traits were highly correlated in our dataset ($R^2$ = 0.85, p < 0.0001, n = 20). Because these two traits did not vary independently in our relatively small sample of 20 hummingbirds, we could not consider them independently. We therefore selected wing length as the more robust measure of wing size, because unlike area, wing length is less prone to measurement error as a result of variation in feather overlap when wings are positioned for measurements. We verified that our results were consistent when using wing area instead of length, and thus these two traits should be considered interchangeable as measures of wing size in this study. Because both wing morphology and muscle capacity may influence burst performance, we used burst performance controlled statistically for wing morphology as a measure of burst muscle capacity. Further details are provided below in the Statistical analysis section.

All procedures were conducted under approval of the Institutional Animal Care and Use Committee at the University of California, Riverside and the Animal Care Committee at the University of British Columbia.

### Tracking system

We used an automated tracking system to measure both body position and orientation of flying birds in three dimensions (*Video 1*). A complete description of the tracking algorithm and hardware components is provided in (*Straw et al., 2011*). The core algorithms were written in Python (*Python Software Foundation, 2012*), and are available via github (PyMVG: https://github.com/strawlab/pymvg; adskalman: https://github.com/astraw/adskalman; MultiCamSelfCal: https://github.com/strawlab/MultiCamSelfCal/). We adapted this system for recording hummingbird solo and competitive flight trajectories with four or five digital cameras (GE680, Allied Vision Technologies, Burnaby, Canada). The cameras were mounted on the ceiling and recorded at 640 x 480 pixel resolution at 200 frames per second (*Figure 1a*). We calibrated the filming volume by moving a

single light-emitting diode throughout the arena to acquire data for an automated self calibration algorithm (*Svoboda et al., 2005*). This algorithm provides a relative calibration (non-linear warping distortion parameters and 3x4 camera calibration matrices) across all cameras. This calibration is brought into absolute terms (the scale, rotation, and translation are found) by matching a manually measured 3D model of the flight arena with reconstructed image coordinates using the 'estsimt' function of the MultiCamSelfCal toolbox (*Svoboda et al., 2005*).

To minimize the effect of errors in the 3D tracking, we used a forward/reverse non-causal Kalman filter (Rauch–Tung–Striebel smoother) applied to the online state estimate of position and velocity from the realtime Kalman filter. The smoothing parameters were chosen so that seven traces of a tracked, falling object yielded an average peak acceleration of 9.8 m/s$^{2.}$ The process covariance matrix we used is:

$$Q_{pos} = \sigma^2 \times \begin{bmatrix} \frac{T^3}{3} & 0 & 0 & \frac{T^2}{2} & 0 & 0 \\ 0 & \frac{T^3}{3} & 0 & 0 & \frac{T^2}{2} & 0 \\ 0 & 0 & \frac{T^3}{3} & 0 & 0 & \frac{T^2}{2} \\ \frac{T^2}{2} & 0 & 0 & T & 0 & 0 \\ 0 & \frac{T^2}{2} & 0 & 0 & T & 0 \\ 0 & 0 & \frac{T^2}{2} & 0 & 0 & T \end{bmatrix}$$

where $\sigma^2$ is 0.01 and $T$ is the interval between frames (0.005 s). The observation covariance matrix we used is:

$$R_{pos} = \begin{bmatrix} 0.000144 & 0 & 0 \\ 0 & 0.000144 & 0 \\ 0 & 0 & 0.000144 \end{bmatrix}$$

*Figure 1c and d* show examples of two trajectories with plots of the unsmoothed data, the data smoothed with $Q_{pos}$ and $R_{pos}$, and the effects of two different smoothing parameters ($R_{pos}$ x 10, $R_{pos}$x 0.1).

Following establishment of the 3D trajectories, the tracking system assigned 3D body orientation vectors to each bird in each frame based on 2D estimates of the long axis of the body. Body orientation was estimated using an algorithm that fit orientations to the body axis in each 2D image. Each sequence of five consecutive images cropped around the bird was aligned at the optical center of intensity. Averaging these images effectively eliminated the wings and emphasized the body. Orientation was estimated by calculating the covariance matrix of the image luminance and then computing the eigensystem of this covariance matrix. The eigenvector associated with the largest eigenvalue was taken as the orientation. Orientation vector assignments were also smoothed with a Kalman filter using more restrictive smoothing parameters than were used to smooth body position. To determine appropriate smoothing parameters we replotted the smoothed body orientation vectors onto a sample of videos, and visually chose the ones that provided the best fit. The process covariance matrix used for body orientation ($Q_{ori}$) is the same as the process covariance matrix used for body position ($Q_{pos}$) and the observation covariance matrix used is:

$$R_{ori} = \begin{bmatrix} 0.00000144 & 0 & 0 \\ 0 & 0.00000144 & 0 \\ 0 & 0 & 0.00000144 \end{bmatrix}$$

Once the body orientations were calculated we used a dynamic programming algorithm to decide which end of the vector was the head and which end was the tail. The direction of the head was chosen based on the direction of the previous orientation, the direction of travel, and the vertical up direction. For each frame ($n$), the 'cost' associated with the two possible orientations ($\vec{Ori}$, -$\vec{Ori}$) were calculated:

$$Cost_{Ori} = Speed\left(\frac{\vec{Ori_n} \cdot \vec{Vel}_{mod}}{\vec{Ori_n}\vec{Vel}_{mod}}\right) + (1 - Speed)\left(\frac{\vec{Ori_n} \cdot \vec{Up}}{\vec{Ori_n}\vec{Up}}\right) + (1 - Speed)\left(\frac{\vec{Ori_n} \cdot \vec{Ori}_{n-1}}{\vec{Ori_n}\vec{Ori}_{n-1}}\right)$$

$$Cost_{-Ori} = Speed\left(\frac{-\vec{Ori_n} \cdot \vec{Vel}_{mod}}{\vec{Ori_n}\vec{Vel}_{mod}}\right) + (1 - Speed)\left(\frac{-\vec{Ori_n} \cdot \vec{Up}}{\vec{Ori_n}\vec{Up}}\right) + (1 - Speed)\left(\frac{-\vec{Ori_n} \cdot \vec{Ori}_{n-1}}{\vec{Ori_n}\vec{Ori}_{n-1}}\right)$$

where $\vec{Ori}$ is the body vector, $\vec{Vel}_{mod}$ is the modified velocity vector tipped up 15° towards the vertical direction. $\vec{Up}$ is the vertical direction vector, $\vec{Ori}_{n-1}$ is the orientation during the previous frame, and if the magnitude of the velocity is greater than 0.5m/s:

$$Speed = \vec{Vel}$$

otherwise:

$$Speed = 0.5$$

This approach accounted for the tendency of hummingbirds to fly forwards and with an upright posture, but allowed for exceptions in the case of backwards flight, inversions, and dives, particularly if these occurred at low speeds.

The magnitudes of calculated accelerations and, to a lesser extent, velocities derived from position data were influenced by the specific smoothing parameters. Examples of maneuvers with different smoothing parameters and their effects on the calculated performance metrics are given in *Figure 1c and d*. This influence of smoothing parameters is a well known limitation of video tracking (*Walker, 1998*). Thus, although acceleration values are comparable within a study, caution must be applied when comparing the magnitude of acceleration values among studies differing in camera frame rate, filming volume, calibrations, and smoothing parameters (*Walker, 1998*). For our final performance metrics we used instantaneous body orientation and orientation velocity, but not orientation acceleration.

The automated tracking system extracted the 3D coordinates of multiple flying animals and saved each trajectory as a separate object (*Video 1*). An object began when the tracking system detected new movement and ended when either the object stopped moving, the error in the 3D reprojection grew too large, or multiple objects came within 2 cm of each other. In our experiments tracking hummingbird flight led to two problems in determining distinct objects. The first is that very stable hovering can be misidentified as perching. For example, as a bird went into an extended hovering bout, such as at a feeder, the tracking system detected the cessation of movement and ended the trajectory. Conversely, when the bird perched at the end of a flight or in between two flights, especially if it continued to move its head or fluff its feathers, the tracking system treated the bird as moving and continued the trajectory. Because our study focused on identifying and analyzing relatively long, moving trajectories, these types of errors did not cause problems. The second challenge concerned identification of birds during close encounters in competition trials. When two tracked objects became close to each other, even if they did not physically touch, the tracking system could not accurately distinguish them. We used a conservative solution and terminated the trajectories whenever two birds came close enough that the tracked objects merged. Birds were later identified manually by a team of digitizers who viewed the videos and assigned each object number to either the marked or unmarked bird.

## Statistical analysis

The automated digitization produced a small number of extreme tracking errors, which we did not want to unduly influence statistical analyses. We accordingly removed values >5 SDs more extreme than the mean for each performance metric. The trimmed values comprised only 0–0.31% of the original pooled sample size for each metric. We next calculated the mean of each performance metric for each bird-trial combination (n= 52 means; 20 birds in 20 solo trials and 16 paired competition trials). All statistical analyses were performed on these bird-trial means using R 3.1.1 (R *Development Core Team, 2014*), and the data used for the analysis are available online (*Segre et al., 2015*).

Repeatability, or the intra-class correlation coefficient (ICC), is defined as the proportion of variation that is attributable to differences among individuals (*Nakagawa and Schielzeth, 2010*). We estimated repeatability for each performance metric from an intercept-only mixed effects model that included estimates of the population intercept (i.e., the grand mean) as well as an individual intercept for each bird (*Nakagawa and Schielzeth, 2010*). Such a model has two variance components, the variance of the random intercept values (variance among individuals) and a residual variance associated with the error term. Repeatability is the variance among individuals divided by the total variance (*Nakagawa and Schielzeth, 2010*). We used parametric bootstrapping with 5000 iterations to obtain confidence intervals for these repeatability estimates via the bootMer function in the lme4 (v1.1.7) package.

Because our second question involved evaluating several possible scenarios for the influence of morphology and burst performance on maneuverability, we used an information-theoretic approach to multi-model inference (*Burnham and Anderson, 2010*). Unlike dichotomous null hypothesis testing, this approach quantifies support for multiple hypotheses, and it avoids the problem of eliminating potentially important predictors when two or more alternative models are equally well supported. The output for interpretation includes the effect size and relative importance of each predictor, and there are no null hypotheses or P values associated with this approach. As a measure of effect size we report the standardized partial regression coefficient, std β, for each predictor, which can be used to compare their independent associations with a given response variable. Unstandardized regression coefficients corresponding to units of the predictor variables are provided in *Supplementary file 1*.

We also examined associations between burst performance, wing size, and wing shape because our load lifting assay may have incorporated effects of wing morphology as well as muscle capacity. The mass of weights lifted during load lifting was not significantly associated with wing length in our sample of 20 individuals (p = 0.87), however, it was negatively associated with wing aspect ratio (p = 0.04) controlling for site. Thus in our model analyses we used residual burst performance controlling for wing aspect ratio and site as a measure of burst muscle capacity independent of a bird's wing morphology.

We considered eight candidate mixed-effects models that could plausibly explain variation in each maneuvering performance metric (*Table 5*). All candidate models included an individual intercept for each bird (the random intercept term) and were fit using the nlme (v 3.1–117) package (*Zuur et al., 2009*). The intercept-only model included an estimate of the population intercept (grand mean) and random intercept terms, but no fixed effects. Other candidate models are listed in *Table 5*. All models except the intercept-only model included the fixed effects of competitor presence, body mass, and experiment to account for potential effects of these factors. Experiment had three levels, one for each round of trials (California 2009, 2010, British Columbia 2014) to account for differences such as location, time of year, and filming conditions.

Two issues arose in the preliminary examination of data. The first issue was that five of the performance metrics were significantly influenced by the number of days a bird had been in captivity. We therefore included an additional fixed effect of the number of days since capture when analyzing these five metrics (*Table 5*). The second issue was that one of the metrics, the heading change in pitch-roll turns ($PRT_{deg}$), had three values that were significant outliers (Grubb's test, all G > 3.09, all p<0.03; *Figure 7*). We determined that these three statistical outliers were not errors in the tracking system but were instead derived from one individual that used pitch-roll turns to make small heading changes, unlike the other birds. We omitted these outliers from the analysis of heading change in pitch-roll turns to ensure that all fitted Gaussian models met the required assumptions, with no other

**Table 5.** Candidate models of maneuvering performance. All models include an intercept as well as a random effect of bird identity to account for repeated measures of individuals.

| Model | Fixed effects | Description |
|---|---|---|
| 1. | Solo/comp + experiment + body mass + wing length | Wing size |
| 2. | Solo/comp + experiment + body mass + wing aspect ratio | Wing shape |
| 3. | Solo/comp + experiment + body mass + wing length + wing aspect ratio | Wing size & shape |
| 4. | Solo/comp + experiment + body mass + weight lifted | Burst power |
| 5. | Solo/comp + experiment + body mass + weight lifted + wing length | Burst power & wing size |
| 6. | Solo/comp + experiment + body mass + weight lifted + wing aspect ratio | Burst power & wing shape |
| 7. | Solo/comp + experiment + body mass + weight lifted + wing length + wing aspect ratio | Burst power, wing size & shape |
| 8. | Intercept-only | |

*Candidate models 1-7 also include a fixed effect of days post-capture for the following metrics: $Vel_{max}$, $AccHor_{max}$, $DecHor_{max}$, $Arc_{vel, \, avg}$, and $Arc_{cent, \, max}$

outliers or problems of skew or heteroskedasticity. The best-fit model for heading change in pitch-roll turns was the intercept-only model regardless of whether the outliers were included.

To quantify the variance explained by the fixed effects of interest in each model, we calculated the marginal $R^2_{GLMM(m)}$ using the r.squaredGLMM function in the MuMIn (v1.10.5) package (**Nakagawa and Schielzeth, 2013**). This measure does not have all the properties of a traditional coefficient of determination, but like $R^2$ it ranges from 0 to 1, and it is an appropriate estimate of the variance explained by the fixed effects in a mixed model. We removed the effect of experiment and the number of days post-capture when calculating $R^2_{GLMM(m)}$, because these were not effects of interest. Thus, $R^2_{GLMM(m)}$ provides a measure of the variance explained by the other supported fixed effect variables.

We evaluated the support for different models using the Akaike information criterion (AICc) adjusted for small sample sizes. This was calculated using the MuMIn (v 1.10.5) package with maximum likelihood estimation. We defined the group of supported models as those with a difference in AICc < 2 from the best-fit model for each performance metric. If no other models came within 2 AICc units of the best-fit model, we present effect size measures, their confidence intervals, and $R^2_{GLMM(m)}$ for only that model. Otherwise, we present averages of all supported models. Details of all candidate models are provided in **Supplementary file 1**.

Our third question concerned the influence of competitor presence on the performance metrics. If the confidence interval for the coefficient estimate of competitor presence excluded zero, we examined the magnitude and direction of that effect. Positive coefficient estimates indicate that performance was higher during competitive flights, whereas negative coefficients indicate that performance was lower in the presence of a competitor.

## Acknowledgements

Adam Behroozian and Tyson Read assisted with data collection. Tungesh Kapil, Janet Li, Sachiko Ouchi, Jordan Roth, Sorosh Safa, Humraaz Samra, Nandhini Sankhyan, Tom Tsou, Sherry Young, Bo Zhang assisted with behavioral scoring.

## Additional information

### Funding

| Funder | Grant reference number | Author |
|---|---|---|
| National Science Foundation | IOS 0923849 | Douglas L Altshuler |
| Natural Sciences and Engineering Research Council of Canada | 402667 | Douglas L Altshuler |

| National Science Foundation | IOS 0923802 | Michael H Dickinson |

The funders had no role in study design, data collection and interpretation, or the decision to submit the work for publication.

## Author contributions

PSS, DLA, Conception and design, Acquisition of data, Analysis and interpretation of data, Drafting or revising the article; RD, Conception and design, Analysis and interpretation of data, Drafting or revising the article; VBZ, MHD, Conception and design, Drafting or revising the article; ADS, Conception and design, Drafting or revising the article, Contributed unpublished essential data or reagents

## Author ORCIDs

Paolo S Segre, http://orcid.org/0000-0002-2396-2670
Andrew D Straw, http://orcid.org/0000-0001-8381-0858
Douglas L Altshuler, http://orcid.org/0000-0002-1364-3617

## Ethics

Animal experimentation: All procedures were conducted under approval of the Institutional Animal Care and Use Committee at the University of California, Riverside and the Animal Care Committee at the University of British Columbia.

# Additional files

## Supplementary files

• Supplementary file 1. Details and ranking of candidate models of hummingbird maneuvering 788 performance, and unstandardized regression coefficients.

## Major datasets

The following datasets were generated:

| Author(s) | Year | Dataset title | Dataset URL | Database, license, and accessibility information |
| --- | --- | --- | --- | --- |
| Paolo S Segre, Roslyn Dakin, Victor B Zordan, Michael H Dickinson, Andrew D Straw, Douglas L Altshuler | 2015 | Data from: Burst muscle performance predicts the speed, acceleration, and turning performance of hummingbirds | http://dx.doi.org/10.5061/dryad.14762 | Available at Dryad Digital Repository under a CC0 Public Domain Dedication |

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
