## [Decision Letter]

Thank you for submitting your work entitled "Burst muscle performance predicts the speed, acceleration, and turning performance of hummingbirds" for consideration by *eLife*. Your article has been reviewed by three peer reviewers, and the evaluation has been overseen by a Reviewing Editor and Eve Marder as the Senior Editor.

The reviewers have discussed the reviews with one another and the Reviewing editor has drafted this decision to help you prepare a revised submission.

Summary:

The reviewers agree in general that this is an interesting and well done study as summarized by one reviewer:

"This study of hummingbird flight is the first to parse out the effects of wing shape vs. muscle capacity in regard to maneuverability. The study system and methodology are ideal for addressing this question, and the manuscript convincingly demonstrates the dominating effect of muscle physiology on maneuverability. Quite interesting. This is supported by a substantial dataset (collected across multiple years), with rigorous analyses and writing that is a pleasure to read. Overall, we believe that this paper will make a strong impact on the field of animal flight mechanics."

That said, all the reviewers had comments provided below. In particular there is concern that you have corrected for body weight twice. Please address all the issues in responding to the reviews with a clear indication of how you have responded.

Comments:

Subsections “Tracking System” and “Maneuvering performance metrics”: General comments on the use of the terms azimuth, pitch, and yaw. Azimuth is a global coordinate reference; pitch and yaw are traditionally body coordinate. Body axis orientation taken from the 2d ellipsoidal trace gives long axis (head/tail) in a local (body) coordinate space; we take it the other two orthogonal axes were assigned without anatomical reference (body lateral and body dorso-ventral). This reference would be required to determine body coordinate space definitions of pitch and yaw (and roll) – which are the traditional uses of the terms (yaw = rotation around a dorso-ventral body axis; pitch = rotation around lateral axis etc.). We suggest the authors make it clear (in the main text) that the "pitch" is therefore not necessarily rotation around the lateral axis (say glenoid to glenoid) of the body, but rather a global "pitch", with the lateral axis being global horizontal (i.e., orthogonal to gravity). Thus, making inferences regarding the anatomical mechanisms in play during these rotations is difficult; a hummingbird in a 90-degree bank to the left, but [body] yawing right will be producing a global "pitch". This problem is in part addressed but not entirely resolved.

Subsection “Maneuvering performance metrics”, third paragraph: 10 cm ~ 1 body length. Is this why 10 cm was chosen? What would the translational velocity cutoff then be? Rather important; 10 cm of movement might be fairly high velocity at some sampling rate. Reading on – the definition of "arcing turn", which has a >. 5 m/s and refers again to a 10 cm limit for vertical distance traveled. Do these reflect the same cutoffs?

Subsection “Maneuvering performance metrics”, fifth paragraph: Yes, if the body is purely vertical, azimuth change is through roll rotation, which is not measured (ambiguous local body coordinates). An azimuth change for a horizontal body will be purely a result of [local coordinate] yaw, provided the animal is not banked; if it is banked, say 45 degrees, then azimuth change will be a result of both [local coordinate] pitch and yaw; at a ninety degree bank, azimuth change is entirely a result of local coordinate pitch. we doubt it changes the overall statistical inferences, but the lack of unambiguous anatomical references may change the magnitude of the accelerations observed and the mechanisms employed to affect those changes – and that's some of what this paper is about.

For a hummer in a steep bank, the rotational acceleration in the azimuth will be affected by largely local coordinate pitch changes, which are a product of bilaterally symmetrical force production of the wings. With no bank, the rotational acceleration in azimuth will be a result of asymmetrical force production by the wings; given the moments of inertia for both these rotations is the same (the radius of gyration for both is the long axis of the body), the accelerations will be smaller for these pure no-bank all-yaw azimuth turns. (Although should the bird produce an posteriorly-directed upstroke force while producing a forward directed downstroke… the Bobcat Loader, or Sherman Tank turn.)

At any rate, while there is probably precedent for use of the terms yaw and pitch in a global sense, it think it's important to be specific here, especially given this paper is making some inferences regarding the anatomical mechanisms used to maneuver.

Subsection “Maneuvering performance metrics”, seventh paragraph: Again, naming it "pitch-roll" further suggests you know around which body axis these maneuvers occur.

Discussion, seventh paragraph, and throughout: I'm concerned that the relative lack of effect of morphology on influence performance may be because the wrong morphology was examined. Maneuvering accelerations are the result of the forces generated relative to the inertia of the bird's mass. Forces are proportional to wing velocity and area (not wing aspect ratio or length – used alone in fixed effect models 1 & 5). Simultaneous effects – significantly negative coefficient for mass, positive for length, and negative for aspect ratio – would infer a wing loading effect, but might it get statistically buried? We think rooting these statistical hypotheses more firmly and clearly in Newtonian expectations would be wise. For example: the biggest effect seen for Acc centripetal max is wing shape. Yes, there may be unsteady effects here, but we know of no aerodynamic theory predicting how aspect ratio would strongly affect this performance variable, given how it was defined.

Wing loading (mass/wing area) would be a more straightforward variable to include, or just area.

A few minor issues: The most salient of these is that it appears to me the authors may have corrected for body mass twice when only one correction was necessary, potentially reducing the size of effect of other parameters. Whether or not this occurred depends on the exact statistical models used and we suggest that the authors examine their models and explain the logic with respect to body mass with a small addition to the text even if no corrections are required.

If we read things correctly this work accounts for body mass in creating the burst performance metric derived from load lifting performance and then also including an intercept for body mass in the maneuvering performance metrics, but from the tables it seems that body mass is never an important effect with a CI not including zero. Is this because it is already corrected for it once and if so, why is a second correction included? The double-correction seems most curious in the acceleration data since load lifting and accelerations are both dependent on force. Perhaps it is simplest to leave body mass out of the burst performance correction and include it in each of the maneuver models? In any case, please provide some explanation of the mass normalization logic in the manuscript text.

The study is flawed overall by the effect that cage size has on the hummingbird flights as compared to actual outdoor flight performance, but this is noted by the authors and outside the scope of what is correctable, given their dataset. We think the results are nevertheless informative and interesting.

Abstract: morphology or physiological -> morphology and physiological, at least according to the Warrick paper you cite later.

Subsection “Maneuvering performance metrics”, fifth paragraph: azimuthal rotation is implemented by rolling the body axis -> rolling about the body axis.

Introduction, first paragraph: Maneuverability is first mentioned here and is a main subject of the paper, but is never defined. We would suggest including it somewhere, and following Dudley's definitions (2002, Int. Comp. Biol.).

Subsection, “Animals and experimental trials”, second paragraph: "We recorded a two-hour solo trial for each bird." It would be worth noting that the recording was with high-speed video.

Subsection “Animals and experimental trials”, third paragraph: "Measurements of wing length and aspect ratio were calculated using custom analysis software in MATLAB". Please tell us how the metrics were defined and generally calculated, for point of comparison with future studies.

Subsection “Tracking System”, first paragraph: "The filming volume was calibrated by moving a single light-emitting diode throughout the arena”. The volume couldn't be calibrated per se by a waved light; was this used for tie points?

In the same subsection, you say: "To minimize the effect of errors in the 3D tracking, we used a forward/reverse non-causal Kalman filter (Rauch-Tung-Striebel smoother)." Applied to what?

Still regarding the same subsection of the text, how were velocities and accelerations calculated? Are they an output from the filter? Where the process covariance matrices are shown, what is the vector multiplied to Q? It seems that estimation for the velocities was included in the filter, with positions being measured. Please clarify the details.

Subsection “Tracking System”, second paragraph and Figure 1: How does the choice in smoothing parameter affect body orientation?

In the same subsection of the text, you state: "Thus, although acceleration values are comparable within a study, caution must be applied when comparing the magnitude of acceleration values among studies differing in camera frame rate, filming volume, calibrations, and smoothing parameters." I believe that Walker (1998) made this same point (and so probably should be cited).

Also in “Tracking System”: What method/function was used to fit the ellipse?

Figure legends:

Figure 1. "The trajectory presented in B is a 2D view of the trajectory shown in A." Is it the top view, x-y projection? You also state: "Level of smoothing had little effect on the performance metrics measured from the maneuvers." The smoothed accelerations range from 10 to 15 m/s2, compared to the unsmoothed value of 54. So although this statement is strictly true, the smoothing did have a large effect on the reported values.

Figure 4. "Aspect ratio was associated with four maneuvering performance metrics." Only 2 metrics are shown. Why not the other 2?

Figure 5. Same comment, 5 vs. 4 metrics shown.

Figure 7. Add "(Arc)" and "(PRT)" after their spelled-out versions.

---

## [Author Response]

*Subsections “Tracking System” and “Maneuvering performance metrics”: General comments on the use of the terms azimuth, pitch, and yaw. Azimuth is a global coordinate reference; pitch and yaw are traditionally body coordinate. Body axis orientation taken from the 2d ellipsoidal trace gives long axis (head/tail) in a local (body) coordinate space; we take it the other two orthogonal axes were assigned without anatomical reference (body lateral and body dorso-ventral). This reference would be required to determine body coordinate space definitions of pitch and yaw (and roll) – which are the traditional uses of the terms (yaw = rotation around a dorso-ventral body axis; pitch = rotation around lateral axis etc.). We suggest the authors make it clear (in the main text) that the "pitch" is therefore not necessarily rotation around the lateral axis (say glenoid to glenoid) of the body, but rather a global "pitch", with the lateral axis being global horizontal (i.e., orthogonal to gravity). Thus, making inferences regarding the anatomical mechanisms in play during these rotations is difficult; a hummingbird in a 90-degree bank to the left, but [body] yawing right will be producing a global "pitch". This problem is in part addressed but not entirely resolved.*

We have followed the reviewers’ suggestion and added these sentences to the subsection: “Because the video tracking system did not allow a measurement of body roll, we decided to use a global coordinate system instead of a body axis-centered coordinate system. In our frame of reference, pitch is a global measure defined relative to the horizontal plane.”

*Subsection “Maneuvering performance metrics”, third paragraph: 10 cm ~ 1 body length. Is this why 10 cm was chosen? What would the translational velocity cutoff then be? Rather important; 10 cm of movement might be fairly high velocity at some sampling rate. Reading on – the definition of "arcing turn", which has a >. 5 m/s and refers again to a 10 cm limit for vertical distance traveled. Do these reflect the same cutoffs?*

We have added the following sentence to address this concern: “We chose 10 cm as a general cutoff here and elsewhere because this value is close to the body length of a bird and the wing span at mid-downstroke, thus providing a good threshold for distinguishing translational motion.”

*Subsection “Maneuvering performance metrics”, fifth paragraph: Yes, if the body is purely vertical, azimuth change is through roll rotation, which is not measured (ambiguous local body coordinates). An azimuth change for a horizontal body will be purely a result of [local coordinate] yaw, provided the animal is not banked; if it is banked, say 45 degrees, then azimuth change will be a result of both [local coordinate] pitch and yaw; at a ninety degree bank, azimuth change is entirely a result of local coordinate pitch. We doubt it changes the overall statistical inferences, but the lack of unambiguous anatomical references may change the magnitude of the accelerations observed and the mechanisms employed to affect those changes – and that's some of what this paper is about.*

*For a hummer in a steep bank, the rotational acceleration in the azimuth will be affected by largely local coordinate pitch changes, which are a product of bilaterally symmetrical force production of the wings. With no bank, the rotational acceleration in azimuth will be a result of asymmetrical force production by the wings; given the moments of inertia for both these rotations is the same (the radius of gyration for both is the long axis of the body), the accelerations will be smaller for these pure no-bank all-yaw azimuth turns. (Although should the bird produce an posteriorly-directed upstroke force while producing a forward directed downstroke… the Bobcat Loader, or Sherman Tank turn). At any rate, while there is probably precedent for use of the terms yaw and pitch in a global sense, it think it's important to be specific here, especially given this paper is making some inferences regarding the anatomical mechanisms used to maneuver.*

We hope that our clarifications of pitch, yaw, and roll (see additions in response to comments 1 and 4) have addressed these concerns.

*Subsection “Maneuvering performance metrics”, seventh paragraph: Again, naming it "pitch-roll" further suggests you know around which body axis these maneuvers occur.*

We have added the following clarifying statements to the relevant section: “Just as we did for the yaw turns, we assume that above a pitch angle of 75°, the rotation is primarily dominated by a body axis roll, even if there may be a slight yawing component. For this reason, we maintain the established 'pitch-roll' terminology to describe these types of turns.”

*Discussion, seventh paragraph, and throughout: I'm concerned that the relative lack of effect of morphology on influence performance may be because the wrong morphology was examined. Maneuvering accelerations are the result of the forces generated relative to the inertia of the bird's mass. Forces are proportional to wing velocity and area (not wing aspect ratio or length – used alone in fixed effect models 1 & 5). Simultaneous effects – significantly negative coefficient for mass, positive for length, and negative for aspect ratio – would infer a wing loading effect, but might it get statistically buried? We think rooting these statistical hypotheses more firmly and clearly in Newtonian expectations would be wise. For example: the biggest effect seen for Acc centripetal max is wing shape. Yes, there may be unsteady effects here, but we know of no aerodynamic theory predicting how aspect ratio would strongly affect this performance variable, given how it was defined.*

We have made several changes to the manuscript to the address this concern, and the most substantial is the new second paragraph of the Discussion:

“Why were body mass and wing size not associated with maneuvering performance? […] Greater within-species variation in wing morphology, and to assess maneuverability across different hummingbird species with divergent morphologies.”

To further acknowledge that the results of this analysis may be constrained by the lack of wing morphology variation in the species under consideration, we have added the common species name (“Anna’s”) to the title, which now reads, “Burst muscle performance predicts the speed, acceleration, and turning performance of Anna’s hummingbirds”.

*Wing loading (mass/wing area) would be a more straightforward variable to include, or just area.*

We understand the source of this concern. We have added the following sentences to explain our choice of the wing size metric included in our analysis:

“We considered wing area and wing length as two potential measures of wing size, but these traits were highly correlated in our dataset (R^2^ = 0.85, P < 0.0001, n = 20) […] these two traits should be considered interchangeable as measures of wing size in this study.”

Because we now explain that we also considered wing area, we have added the mean and range for this measurement to Table 5.

*A few minor issues: The most salient of these is that it appears to me the authors may have corrected for body mass twice when only one correction was necessary, potentially reducing the size of effect of other parameters. Whether or not this occurred depends on the exact statistical models used and we suggest that the authors examine their models and explain the logic with respect to body mass with a small addition to the text even if no corrections are required.*

*If we read things correctly this work accounts for body mass in creating the burst performance metric derived from load lifting performance and then also including an intercept for body mass in the maneuvering performance metrics, but from the tables it seems that body mass is never an important effect with a CI not including zero. Is this because it is already corrected for it once and if so, why is a second correction included? The double-correction seems most curious in the acceleration data since load lifting and accelerations are both dependent on force. Perhaps it is simplest to leave body mass out of the burst performance correction and include it in each of the maneuver models? In any case, please provide some explanation of the mass normalization logic in the manuscript text.*

We are grateful to the reviewer(s) for catching this error. We should not have analyzed burst performance relative to body mass because we had already included body mass as an independent variable. We have revised our analysis, and this change did not affect any of our conclusions because body mass is not related to either burst performance or maneuvering performance in this data set. There were very small changes to some values. For example, in Table 3, row 1, for the model of velocity, the beta coefficient for body mass [95% CI] changed from 0.09 [0.002,0.18] to 0.09 [0.001,0.18]. An example of a larger change is in Table 3, row 2, for the model of horizontal acceleration, the beta coefficient for body mass [95% CI] changed from 0.13 [-0.34,0.59] to 0.20 [-0.28,0.69]. Even here, this did not change our conclusion because of the broad range including zero. We have corrected the relevant tables and figures, and a few places in the text where we provide effect sizes.

*The study is somewhat flawed by the effect that cage size has on the hummingbird flights as compared to actual outdoor flight performance, but this is noted by the authors and outside the scope of what is correctable, given their dataset. We think the results are nevertheless informative and interesting.*

*Abstract: morphology or physiological -> morphology and physiological, at least according to the Warrick paper you cite later.*

Done.

*Subsection “Maneuvering performance metrics”, fifth paragraph: azimuthal rotation is implemented by rolling the body axis -> rolling about the body axis.*

Done.

*Introduction, first paragraph: Maneuverability is first mentioned here and is a main subject of the paper, but is never defined. We would suggest including it somewhere, and following Dudley's definitions (2002, Int. Comp. Biol.).*

This has been added to the first sentence of the Introduction, which now reads, “The ability of an animal to change the speed and direction of movement, defined as maneuverability (Dudley 2002), can determine its success at avoiding predators, obtaining food, and performing other behaviors that determine the margin between life and death (Webb 1976, Hedenström and Rosén 2001, Walker et al. 2005).”

*Subsection, “Animals and experimental trials”, second paragraph: "We recorded a two-hour solo trial for each bird." It would be worth noting that the recording was with high-speed video.*

Done.

*Subsection “Animals and experimental trials”, third paragraph: "Measurements of wing length and aspect ratio were calculated using custom analysis software in MATLAB". Please tell us how the metrics were defined and generally calculated, for point of comparison with future studies.*

We provided this information in the Methods. The relevant section now reads:

“Immediately following load lifting, we weighed the birds and photographed both wings in an outstretched position against white paper with a reference scale (Chai and Dudley 1995) […] Values for aspect ratio, wing area, and wing length were then calculated based on equations in Ellington (Ellington 1984).”

*Subsection “Tracking System”, first paragraph: "The filming volume was calibrated by moving a single light-emitting diode throughout the arena". The volume couldn't be calibrated per se by a waved light; was this used for tie points?*

We have addressed this concern by modifying the relevant section, which now reads:

“We calibrated the filming volume by moving a single light-emitting diode throughout the arena to acquire data for an automated self-calibration algorithm (Svoboda et al. 2005) […] 3D model of the flight arena with reconstructed image coordinates using the ‘estsimt’ function of the MultiCamSelfCal toolbox (Svoboda et al. 2005).”

*In the same subsection, you say: "To minimize the effect of errors in the 3D tracking, we used a forward/reverse non-causal Kalman filter (Rauch-Tung-Striebel smoother)." Applied to what?*

We extended this sentence to "To minimize the effect of errors in the 3D tracking, we used a forward/reverse non-causal Kalman filter (Rauch–Tung–Striebel smoother) applied to the online state estimate of position and velocity from the realtime Kalman filter."

Still regarding the same subsection of the text, how were velocities and accelerations calculated? Are they an output from the filter? Where the process covariance matrices are shown, what is the vector multiplied to Q? It seems that estimation for the velocities was included in the filter, with positions being measured. Please clarify the details.

We have added information addressing the first two questions to subsection “Maneuvering performance metrics”. The relevant sentences now read:

“Translational velocity and acceleration were calculated by taking the first and second derivatives of an interpolation spline fit to the body position data […]. We took the first derivatives to obtain azimuth and pitch velocities.”

Regarding the queries about the covariance matrices, we now present the covariance matrices in a more conventional format, specifically in terms of T (the interframe interval) and *σ^2^* (a single scalar covariance parameter). These changes were made to [Supplementary-material SD1-data].

Subsection “Tracking System”, second paragraph and Figure 1: How does the choice in smoothing parameter affect body orientation?

We have addressed this in the caption (see below) and in the main text. In the latter case, we added the following sentences to the end of the subsection:

“The magnitudes of calculated accelerations, and to a lesser extent velocities, derived from position data were influenced by the specific smoothing parameters […]. Details of the specific approach for tracking multiple hummingbirds are provided in [Supplementary-material SD1-data]”.

In the same subsection of the text, you state: "Thus, although acceleration values are comparable within a study, caution must be applied when comparing the magnitude of acceleration values among studies differing in camera frame rate, filming volume, calibrations, and smoothing parameters." I believe that Walker (1998) made this same point (and so probably should be cited).

Done.

*Also in “Tracking System”: What method/function was used to fit the ellipse?*

We regret the poor phrasing here. We did not fit an ellipse but instead determined the orientation and position on an elliptical object. We have modified the relevant section, which now reads: “Orientation was estimated by calculating the covariance matrix of the image luminance and then computing the eigensystem of this covariance matrix. The eigenvector associated with the largest eigenvalue was taken as the orientation.”

Figure legends:

Figure 1. "The trajectory presented in B is a 2D view of the trajectory shown in A." Is it the top view, x-y projection?

Yes, and this has now been specified in the figure legend.

You also state: "Level of smoothing had little effect on the performance metrics measured from the maneuvers." The smoothed accelerations range from 10 to 15 m/s2, compared to the unsmoothed value of 54. So although this statement is strictly true, the smoothing did have a large effect on the reported values.

We have removed this sentence and added the following sentences to the figure legend:

“The chosen smoothing parameters for body position were determined by tracking multiple dropped objects and calibrating the Z-axis acceleration to gravity […] However, the level of smoothing for body orientation had minimal effect on the average yaw velocity.”

*Figure 4. "Aspect ratio was associated with four maneuvering performance metrics." Only 2 metrics are shown. Why not the other 2?*

This was an error in the caption, which we have now corrected.

*Figure 5. Same comment, 5 vs. 4 metrics are shown.*

This was an error in the caption, which we have now corrected.

*Figure 7. Add "(Arc)" and "(PRT)" after their spelled-out versions.*

Done.